# Amitriptyline prescribing in public sector healthcare facilities in the Western Cape, South Africa

**Renier Coetzee**[1]*, **Yasmina Johnson**[2], **Johan van Niekerk**[3‡], **Mosedi Namane**[4‡]

**1** School of Pharmacy, University of the Western Cape, Cape Town, South Africa, **2** Western Cape Government Health, Cape Town, South Africa, **3** Khayelitsha Eastern Sub-structure, Metro Health Services, Department of Health, Western Cape Government, Cape Town, South Africa, **4** Vanguard Community Health Centre and School of Public Health and Family Medicine, University of Cape Town, Cape Town, South Africa

☯ These authors contributed equally to this work.
‡ These authors also contributed equally to the work.
* recoetzee@uwc.ac.za

**Data Availability Statement:** All relevant data are within the paper and its Supporting Information files.

**Funding:** The authors received no specific funding for this work.

## Abstract

### Background

Inappropriate medication use is a major patient safety concern, especially for the elderly population. Amitriptyline is widely used in primary care in South Africa and a cross-sectional study found that amitriptyline was prescribed potentially inappropriately in 6.5% of elderly patients. An analysis of prescriptions from the Chronic Dispensing Unit in the Western Cape revealed that amitriptyline was one of the most common medicines prescribed without a suitable diagnosis listed on the prescription.

### Objective

The main objective of the medicine use evaluation (MUE) was to determine whether amitriptyline was prescribed in accordance with recommendations from standard treatment guidelines (STG) and essential medicines lists (EML) endorsed by the National Department of Health, South Africa.

### Methods

A retrospective, cross-sectional, multicentre review of patients' clinical notes was conducted. The study population was selected by systematic random sampling from adult outpatients who were prescribed amitriptyline for longer than three months. Criteria for evaluation included amitriptyline indication and total daily dose prescribed.

### Results

Of the sample of 2237 patient medical records reviewed, 1732 (77.4%) included amitriptyline prescriptions that were according to the approved STG indications. For the approved STG indications, amitriptyline was prescribed mainly for osteoarthritis (25.8%), neuropathies (18.5%) and chronic non-cancer pain (17.9%). Major depressive disorders constituted

**Competing interests:** The authors have declared that no competing interests exist.

only 8.6% of the patient records reviewed; however, doses were atypically low. The main inappropriate indication for amitriptyline was sleep disorders (16%).

## Conclusion

This MUE has highlighted the need to improve the use of amitriptyline in specific patient populations, e.g. the elderly and patients with sleeping disorders.

## Background

Access to essential medicines is critical in addressing health problems and improving the quality of lives of millions of patients around the world. Medicines form an indispensable component of any healthcare system in the prevention, diagnosis and treatment of diseases and in alleviating disability and functional deficiency. Essential medicines are defined as "medicines that satisfy the priority healthcare needs of the population," with the concept being that these medicines are intended to be available at all times, in adequate amounts, in appropriate dosage forms, with assured quality and at a price that the individual and country can afford.[1] The World Health Organization (WHO) further states that rational use of medicines require that "patients receive medication appropriate to their clinical needs, in doses that meet their own individual requirements, for an adequate period of time, and at the lowest cost to them and their community".[2]

Taking the above two concepts into consideration, the prescribing of medicines should generally follow these steps: (a) Determine the diagnosis of the patient; (b) Consider effective and safe treatment options; (c) Select appropriate medicines, which include dose, frequency and duration; (d) Write a clear prescription; (e) Provide patients with the necessary information or counselling; and (f) Follow up to evaluate treatment responses.[3] However, prescribing patterns do not always conform to these ideals, resulting instead in inappropriate and irrational prescribing.[4]

The impact of irrational medicine use can have varying effects on patients and the risk of adverse drug reactions (ADRs) is increased. In older patients the presence of a comorbidity is a strong predictor of repeat admissions for ADRs, especially if they are managed and treated in the community setting.[5] Irrational prescribing can also expose patients to the possibility of developing drug dependence to certain medicines, such as analgesics and tranquillizers.[6] Not only does inappropriate use of medicines have a negative effect on patients, but it also leads to wastage of scarce resources. The WHO estimated that the appropriate use of medicines can result in about 50%–70% cost-efficiency in medicines expenditure.[7]

Inappropriate medication use is a major patient safety concern, especially for the elderly population.[8] In 1987 and 1992, it was reported that between 17.5% and 23.5% of community-dwelling elderly patients in the United States used at least 1 of 20 potential inappropriate medicines.[9] The most frequently prescribed potentially inappropriate medications were diazepam, propoxyphene, dipyridamole, amitriptyline, and chlordiazepoxide; these medicines accounted for 85% of the outpatient visits involving potentially inappropriate medications.[10] A cross-sectional study, including 103 420 patients aged ≥65 years, that was conducted using a database obtained from a pharmaceutical benefit management company in South Africa found that amitriptyline was prescribed potentially inappropriately in 6.5% (n = 36 509) of elderly patients.[11]

Amitriptyline is widely used in primary care in South Africa as it is listed as an Essential medicine in South Africa. In the Western Cape, data from the provincial medicine warehouse, the Cape Medical Depot (CMD), indicated an annual consumption of more than 600 000 packs (included 28s, 56s, 84s and 168s) of amitriptyline 10mg or 25mg with an expenditure of R3.5million (US$231 500.00) for the period July 2016 to June 2017.[12] An analysis done on prescriptions at the Chronic Dispensing Unit in the Western Cape revealed that amitriptyline was one of the most common medicines prescribed without a diagnosis listed on the prescription.[13] Thus, concerns for inappropriate prescribing, together with the wide use and potential for adverse effects, led to the Provincial Pharmacy and Therapeutics Committee (PTC) of the Western Cape requesting that a review of the use of amitriptyline be conducted. To our knowledge limited to no data exists on medicine use evaluations for mental health medicines in low to middle income countries (LMICs). Our objective was to establish if amitriptyline is prescribed according to the recommendations set out in the Standard Treatment Guidelines [14, 15] developed by the South African Department of Health.

## Method

A retrospective, multicentre review of clinical notes of outpatients was conducted in the public health care sector of the Western Cape. The study population included adult outpatients ($\geq$18 years of age) who received amitriptyline for longer than 3 months during the period 01 April 2016 to 31 July 2017.

The folder review was performed between 27 March 2017 and 31 July 2017. A convenience sample of facilities was selected from health facilities that had at least one fulltime pharmacist and doctor. According to South African legislation, amitriptyline (as a schedule 5 medicine) should be prescribed by a doctor. Facilities with low amitriptyline numbers were excluded. At least 30 folders per facility were selected by random systematic selection and reviewed over a 2 to 4-week period within the study period. The sample size was based on the World Health Organization recommendation of 30 folders per facility [16]. The data was sourced from patients' clinical records using a data collection tool developed by a multi-disciplinary team that included pharmacists and doctors working in the healthcare facilities. A pilot study was done at one facility to determine ease of use of the data collection tool as well as to determine if the required analysis could be done with the variables collected. Data collected included demographic data (gender and age), diagnosis, medication prescribed, comorbidities and amitriptyline dosages. Information collected was captured onto a predesigned Microsoft® Office Excel (Microsoft, USA) spreadsheet, from where analyses were performed. Descriptive statistics were used to report proportions and percentages of the outcomes. P-values $\leq$0.05 were considered significant.

The criteria used for evaluation were diagnosis-related. The criteria were determined by using the Standard Treatment Guidelines (STGs) and Essential Medicines List (EML) for Primary Health Care (2014)[14] and Adult Hospital Level (2015)[15]. These guidelines were in use at the time of the review. Table 1 lists the STGs and EML indications for amitriptyline use.

**Table 1. Indications for amitriptyline according to Standard Treatment Guidelines[14, 15].**

| | | |
|---|---|---|
| 1. Major depressive disorders (MDD) (F32.9) | | |
| 2. Pain associated with: | | |
| Neuropathies (E10.2/E11.2/N08.3) | Rheumatoid arthritis (M05.9/M06.9) | Herpes zoster (B20.3) |
| Cancer, non-cancer / other (R52.9) | Osteoarthritis (M19.9) | Migraine (G43) |

To facilitate data collection, the following non-EML indications were added to the data collection tool: irritable bowel syndrome, anxiety, sleeping disorders and overactive bladder. Diagnoses had to be documented unambiguously (not symptoms) in the medical notes to be accepted. Diagnoses that were not listed on the tool or unclear were included as 'other non-pain indications'. Folders were reviewed independently and discrepancies were resolved amongst the authors.

A threshold refers to the percentage of charts or records that will meet or exceed the established criteria for the medicine. Ideally, this threshold will be 100 percent, but realistically, a smaller percentage will be more appropriate to account for exceptions to routine medicine prescribing. The threshold was set at 99% as amitriptyline is prescribed predominantly at primary healthcare level. It is expected that 99% of prescriptions should comply with the STG & EML with regards to the diagnosis, dosage and process issues. A deviation of 1% was allowed for exceptions. The threshold was jointly decided upon by a multidisciplinary team.

### Ethical considerations

Approval to conduct the study was granted by the Western Cape Pharmacy and Therapeutics Committee, the Western Cape Government Health (WC_201806_021) as well as the University of the Western Cape's Biomedical Research Ethics Committee (BM18/4/5). For the purpose of the folder review informed consent from patients were not required by the approval committees as this study was retrospective, data was anonymized before analysis and there was no direct patient contact.

### Results

Out of 193 health facilities where amitriptyline was prescribed, 102 facilities were excluded due to low amitriptyline numbers (i.e. < 15 prescriptions per week). Thus, out of the 91 facilities, sixty-four facilities (70.3% response rate) contributed to the review of amitriptyline use in the Western Cape. A total of 2237 patient folders were reviewed. These patients received amitriptyline for a period of longer than three months. Metropole facilities, rural facilities and transverse facilities (e.g. tertiary hospitals that serve patients across the province) made up 75%, 17% and 8%, respectively, of the sample. Graph 1 indicates the contribution to the sample by the type of facility. Community Day Centres (CDCs), Community Health Centres (CHCs) and clinics contributed 80% towards the total sample, while hospital outpatients made up 20% of the sample (Fig 1).

Demographic information is shown in Table 2. There were two and a half times more females (1639; 73%) than males (597; 27%) receiving amitriptyline. The average age of the patients was 57 years (SD±13.81). Females were on average 2 years older than males, with the average age being 58 years and 56 years, respectively (p<0021).

For the prescriptions reviewed, 1732 (n = 2237; 77.43%) prescriptions for amitriptyline were according to indications approved in the STGs. Thus, almost a quarter of patients received amitriptyline for indications not listed or approved in the STGs.

As more than one indication was reported for some folder reviews of patients with co-morbidities, a total of 2688 indications for use of amitriptyline were reported in the total sample (n = 2237). The proportion of prescriptions by indication is shown in graph 2 (Fig 2). For the approved STG indications (Table 1), amitriptyline was prescribed mainly for osteoarthritis (25.79%), followed by chronic non-cancer pain (17.88%). Major depressive disorders constituted 8.58%, while all neuropathies combined constituted 18.45% of the patient medical records reviewed. At 16.00%, sleeping disorders made up the highest proportion of patient folder

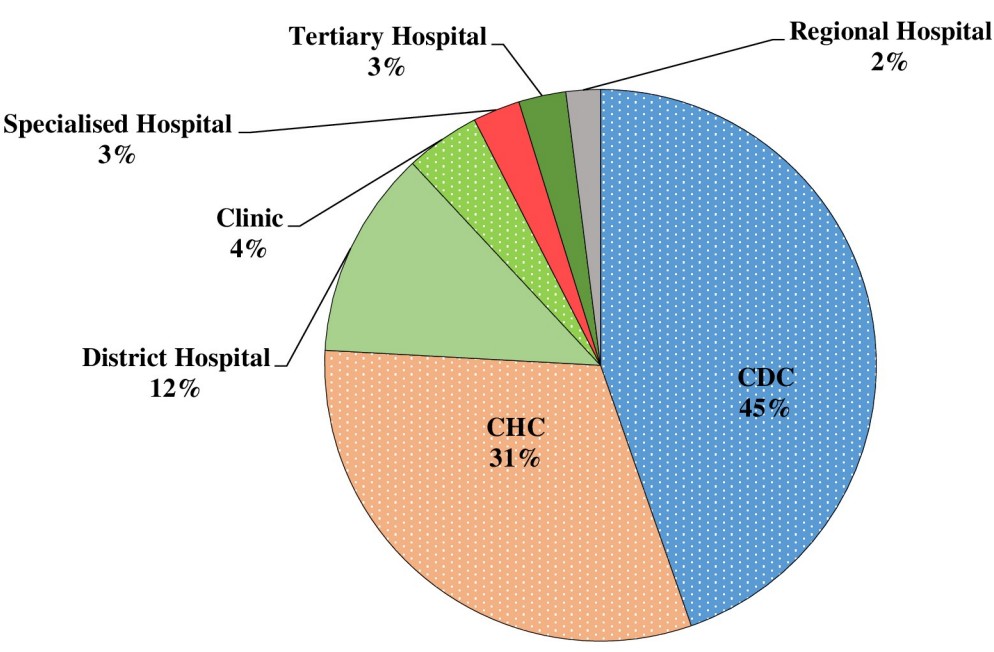

**Fig 1. Proportion of folders reviewed by facility type.**

reviews for a Non-STG indication and this also constituted the indication with the third highest proportion of all folders reviewed.

Table 3 provides information on sex, age and dose per indication. The most common indication within the female sub-group was osteoarthritis/osteoarthrosis (27.9%) and for males it was neuropathies (25.6%). Elderly patients are prone to side effects of amitriptyline. Amitriptyline increases the risk of urinary retention in males over 55 years old, thus the age sub-groups were split into patients that are ≤55 years and patients above 55 years old. The proportion of patients over 55 years were 54%, 55% and 50% for the total sample, females and males, respectively. Doses were split into ≤50mg and >50mg. Around 95% of the doses prescribed for amitriptyline were ≤ 50mg daily; with 86% of these doses being ≤ 25mg daily. For the STG indications (n = 1732), at least 95% of doses were within guideline recommendations. Doses for the Non-STG indications could not be assessed as there were no clinical practice guidelines in place to use as a reference for these indications.

Further analysis showed that for males over 55 years, only 9% of patients in this risk group were on doses greater than 25mg daily. For patients that were ≥ 65 years, doses were generally in the lower range, i.e. 87% of the doses were ≤ 25mg daily and another 10% of doses were > 25mg but ≤ 50mg.

Only 1% of the doses were ≥ 125mg daily and these doses were prescribed for indications such as major depressive disorder (MDD), anxiety disorder (Non-STG indication), sleep

**Table 2. Gender descriptive statistics (n = 2237).**

|  | n | Mean age | Standard deviation | Min age | Max age | Median | Age: CI 95% |
|---|---|---|---|---|---|---|---|
| **Female** | 1639 (73%) | 58 | 13.31 | 18 | 99 | 57 | 57.01–58.30 |
| **Male** | 597 (27%) | 56 | 14.99 | 18 | 93 | 56 | 54.44–56.85 |
| **Total** | 2237* (100%) | 57 | 13.81 | 18 | 99 | 57 | 56.54–57.69 |

*One patient with missing gender

**Table 3. Analysis of sex, age and dose (sub-groups) per indication.**

| Sub-group | Quantity & proportions | EML INDICATIONS | | | | | | | | | NON-EML INDICATIONS | | | | | | |
|---|---|---|---|---|---|---|---|---|---|---|---|---|---|---|---|---|---|
| | | Neuropathy | Post-herpetic neuralgia | Herpes Zoster Ophthalmicus | Rheumatoid arthritis | Osteoarthritis / osteoarthrosis | Chronic cancer pain | Other chronic non-cancer pain | Chronic non-specific pain syndrome | Other pain indication, specify | Major depressive disorder | Anxiety disorders | Sleeping disorders | Irritable Bowel Syndrome | Overactive bladder / Enuresis | Any other non-pain indication, specify | No indication noted in patient folder |
| FEMALES | Females N = 1639 Quantity | 244 | 11 | 3 | 87 | 457 | 44 | 291 | 66 | 150 | 148 | 72 | 265 | 36 | 17 | 102 | 170 |
| | % females | 14.9% | 0.7% | 0.2% | 5.3% | 27.9% | 2.7% | 17.8% | 4.0% | 9.2% | 9.0% | 4.4% | 16.2% | 2.2% | 1.0% | 6.2% | 10.4% |
| | % total sample | 13.9% | 0.5% | 0.1% | 3.9% | 20.4% | 2.0% | 13.0% | 3.0% | 6.7% | 6.6% | 3.2% | 11.8% | 1.6% | 0.8% | 4.6% | 7.6% |
| | >55yrs[@] n = 909 Quantity | 109 | 9 | 1 | 55 | 333 | 27 | 132 | 37 | 74 | 80 | 32 | 175 | 25 | 15 | 61 | 88 |
| | % subgroup[*] | 44.7% | 81.8% | 33.3% | 63.2% | 73.0% | 61.4% | 45.4% | 56.1% | 49.3% | 54.8% | 44.4% | 66.5% | 69.4% | 88.2% | 59.8% | 53.3% |
| | ≤55yrs[@] n = 721 Quantity | 135 | 2 | 2 | 32 | 123 | 17 | 159 | 29 | 76 | 66 | 40 | 88 | 11 | 2 | 41 | 77 |
| | % subgroup[*] | 55.3% | 18.2% | 66.7% | 36.8% | 27.0% | 38.6% | 54.6% | 43.9% | 50.7% | 45.2% | 55.6% | 33.5% | 30.6% | 11.8% | 40.2% | 46.7% |
| MALES | Males n = 597 Quantity | 153 | 2 | 1 | 15 | 120 | 7 | 109 | 26 | 61 | 44 | 27 | 93 | 10 | 2 | 41 | 54 |
| | % males | 25.6% | 0.3% | 0.2% | 2.5% | 20.1% | 1.2% | 18.3% | 4.4% | 10.2% | 7.4% | 4.5% | 15.6% | 1.7% | 0.3% | 6.9% | 9.0% |
| | % total sample | 7.3% | 0.1% | 0.0% | 0.7% | 5.4% | 0.3% | 4.9% | 1.2% | 2.7% | 2.0% | 1.2% | 4.2% | 0.4% | 0.1% | 1.8% | 2.4% |
| | >55yrs n = 299 Quantity | 57 | 1 | 0 | 7 | 91 | 7 | 47 | 14 | 32 | 17 | 10 | 59 | 5 | 0 | 23 | 26 |
| | % subgroup[*] | 37.3% | 50.0% | 0.0% | 46.7% | 75.8% | 100.0% | 43.1% | 53.8% | 52.5% | 38.6% | 37.0% | 63.4% | 50.0% | 0.0% | 56.1% | 48.1% |
| | ≤55yrs n = 298 Quantity | 96 | 1 | 1 | 8 | 29 | 0 | 62 | 12 | 29 | 27 | 17 | 34 | 5 | 2 | 18 | 28 |
| | % subgroup[*] | 62.7% | 50.0% | 100.0% | 53.3% | 24.2% | 0.0% | 56.9% | 46.2% | 47.5% | 61.4% | 63.0% | 36.6% | 50.0% | 100.0% | 43.9% | 51.9% |
| DOSE[#] | Dose ≤50mg n = 2134 Quantity | 383 | 13 | 4 | 97 | 559 | 49 | 378 | 86 | 202 | 171 | 84 | 344 | 45 | 19 | 139 | 216 |
| | % subgroup[*] | 97.0% | 100.0% | 100.0% | 98.0% | 97.2% | 96.1% | 96.9% | 95.6% | 97.1% | 90.0% | 86.6% | 97.5% | 97.8% | 100.0% | 97.2% | 98.2% |
| | Dose >50mg n = 77 Quantity | 12 | 0 | 0 | 2 | 16 | 2 | 12 | 4 | 6 | 19 | 13 | 9 | 1 | 0 | 4 | 4 |
| | % subgroup[*] | 3.0% | 0.0% | 0.0% | 2.0% | 2.8% | 3.9% | 3.1% | 4.4% | 2.9% | 10.0% | 13.4% | 2.5% | 2.2% | 0.0% | 2.8% | 1.8% |

*the proportions are per sub-group per indication

# patients with missing doses were not included in the dose sub-group

@ patients with missing ages were not included in the females-age sub-group

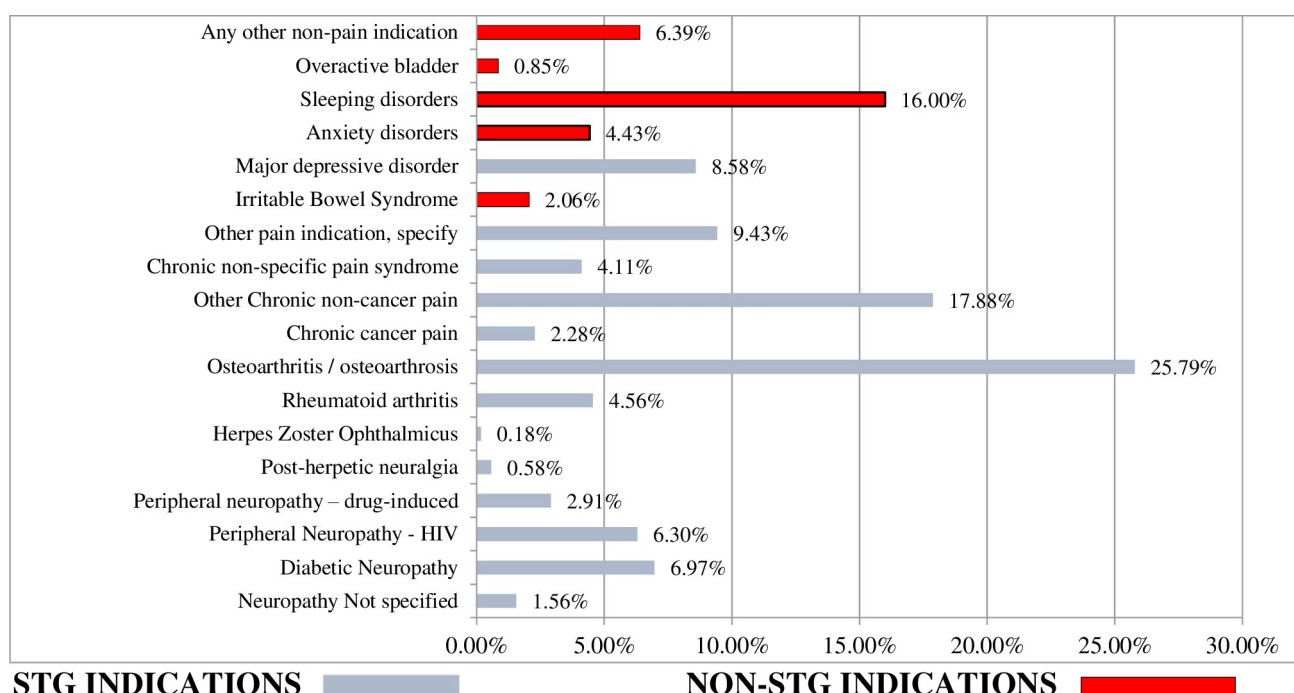

**Fig 2. Proportion of prescriptions by indication of use.**

disorder (Non-STG indication) and osteoarthritis. Although infrequent, high doses (≥125mg daily) of amitriptyline for osteoarthritis were of concern as this was not in line with STG recommendations.

For MDD, 70% of the prescriptions had a dose of ≤ 25mg daily and a further 20% had a dose of 50mg daily. Of these prescriptions, only 41% had another antidepressant or mood stabiliser added, implying that these patients on amitriptyline monotherapy may be sub-optimally managed.

Table 4 indicates amitriptyline prescribed with other pain modulating medicines. Paracetamol was widely used in combination with amitriptyline, with just under two thirds of the prescriptions having this combination. Furthermore, a third of the prescriptions comprised amitriptyline with tramadol. Amitriptyline on its own was predominantly prescribed for sleep

**Table 4. Concomitant pain medicine use.**

| Pain Medicines | Quantity | Percentage |
|---|---|---|
| **Amitriptyline** | | |
| Amitriptyline only | 690 | 30.84% |
| **+ Paracetamol** | | |
| Paracetamol all | 1398 | 62.49% |
| Paracetamol only | 478 | 21.37% |
| **+Ibuprofen/NSAID** | | |
| Ibuprofen/NSAID all | 466 | 20.83% |
| NSAID only | 72 | 3.22% |
| **+Tramadol** | | |
| Tramadol all | 748 | 33.44% |
| Tramadol only | 60 | 2.68% |

Table 5. Thresholds observed according to STG criteria.

| Criteria | Thresholds | Observed |
|---|---|---|
| **Indications** | 99% | 77% |
| **Dosages** | 99% | 95% |
| **Recording of diagnosis information** | 99% | 90% |

disorders and neuropathies. Amitriptyline with paracetamol only (as only other pain-modulating medicine) was predominantly prescribed for neuropathies, osteoarthritis and other chronic non-cancer pain. For amitriptyline with non-steroidal anti-inflammatory drugs (NSAIDs) only or tramadol only, osteoarthritis and other chronic non-cancer pain were the main indications.

Table 5 compares the thresholds set and results observed for the STG indications, appropriate dosing and clear documentation. Observed results of 77%, 95% and 90% were achieved, respectively. The proportion calculated for 'appropriate dosing' was estimated from prescriptions with STG indications only as no recommendation on dosing is provided for Non-STG indications.

## Discussion

With an unknown proportion of the population that would be adherent to guidelines, a proportion of 50% (i.e. $\rho = 0.5$) and a confidence interval of 95% were used to calculate the sample size required i.e. 384. In order to expose all facilities that matched our inclusion criteria to the MUE methodology and requiring at least 30 patient medical record reviews per facility, a total of 2237 patient medical records were reviewed; this was well within the calculated sample size. In the Western Cape, the approach for patient care is that, generally, around 85% of patients should receive healthcare at a primary level, around 12% at hospital level and around 3% at a tertiary care level. Furthermore, the indications for amitriptyline, including follow-up after referral from hospital level, falls mainly within the ambit of primary care. Thus, as 80% of the sample was made up from primary care level patients, this indicated a fair representation of the patient distribution.

In South Africa, the Standard Treatment Guidelines and Essential Medicines List (STGs) are evidence-based medicine recommendations aimed to cater for most of the population [14,15]. Adherence to these recommendations is expected from all public sector healthcare facilities to ensure an effective standard of care, safety and equitable access to medicines. As amitriptyline was used predominantly at primary care according to consumption data, adherence to the STGs was expected to be high; and thus, a high threshold (99%) was set for what is considered appropriate prescribing of amitriptyline in the province.

In our review the majority of the ambulatory patients received amitriptyline as per STGs, mainly as an adjunct for pain control in chronic pain conditions. However, almost one quarter of the prescriptions were for indications that are not in line with the STGs and thus, there is room for improvement. Around 16% of the patients were on amitriptyline for sleep disorders and this may suggest a need for appropriate sleep disorder management opportunities within the public sector. According to the STGs, sleep hygiene and stimulus control are the first steps of treatment; and if medication is required, short-term benzodiazepines are the preferred agents to use for the treatment of insomnia. Non-benzodiazepines tend not to be abused in comparison to benzodiazepines, are effective in insomnia and may offer a better safety profile over benzodiazepines. However, the use of amitriptyline for insomnia is regarded as off-label use. Low-dose amitriptyline has been used successfully in patients with insomnia, but is contraindicated in suicidal patients or in those with cardiac risk factors. It is generally

recommended that antidepressants be prescribed at therapeutic doses when insomnia coexists with a mood disorder.[17] The STGs, therefore, recommend referral of patients that cannot be managed utilising cognitive behaviour methods or short-term benzodiazepines.

The use of low dose amitriptyline in major depressive disorders raised concerns for possible inappropriate management of these conditions. Low doses of amitriptyline for treating mood disorders may not only be ineffective but could also produce unwarranted adverse effects in patients. The lifetime prevalence for major depression in South Africa is 9.8%[18]. In a study conducted in the private healthcare sector of South Africa, only 26.2% of patients who received amitriptyline-containing products were compliant[19]. This raises further concerns when taking under dosing, as seen in our study, into consideration. Ineffective treatment of mood disorders has a substantial risk for suicide.

A third of the patients were on tramadol and amitriptyline, concurrently, and this combination can induce constipation, precipitate urinary retention in elderly men, lower the seizure threshold and cause serotonin syndrome. The current STGs do not provide any recommendation on education or precautions on the potential risk of serotonin syndrome. This review highlights the potential risk of serotonin syndrome in this population.

The highest proportion of prescriptions was indicated for osteoarthritis (26%); however, there is limited evidence for recommending amitriptyline therapy for inflammatory arthritis and some evidence of benefit in fibromyalgia, but no evidence of benefit of amitriptyline in osteoarthritis.[20] Also, in a systematic review to determine effectiveness of amitriptyline in musculoskeletal pain and improved functionality, it was concluded that amitriptyline may be effective, but further research is needed to establish efficaciousness and the specific indication (s) for amitriptyline.[21] A Cochrane review is underway, currently, to determine the safety and efficacy of antidepressants for osteoarthritis.[22] Thus, due to safety concerns and a paucity of evidence of efficacy of amitriptyline in osteoarthritis, amitriptyline was not included in the South African Primary Health Care STG for this indication. Preferably, amitriptyline should therefore not be initiated at primary care level for this indication, especially, because of potential cardiotoxicity in patients with co-morbid cardiovascular conditions. Amitriptyline, however, remains listed in the Adult Hospital STG for osteoarthritis as adjunctive therapy for pain control as this is considered standard of care, currently, and there is uncertainty as to whether amitriptyline is not efficacious[15].

The mean age of the sample was 57 (±14) years and older patients often have multiple diseases, especially co-morbid cardiovascular conditions, requiring multiple medicines; this increases the potential for using inappropriate medicines and ultimately results in various adverse effects. In elderly patients there exist concerns with the use of anticholinergic drugs due to potential confusion, constipation, urination problems, blurry vision and low blood pressure. This review highlighted the importance of considering the benefit (effectiveness) versus the risks when prescribing amitriptyline, especially due to the patient groups involved.

## Conclusion

This review has highlighted the importance of evaluating patients' need for amitriptyline when receiving it for chronic use. In these public healthcare facilities, there is a need to improve the use of amitriptyline in specific patient populations, e.g. elderly men, patients with major depressive disorders, patients with sleeping disorders or with co-morbid cardiovascular conditions.

## Supporting information

**S1 File. De-identified data set.**
(XLSX)

## Acknowledgments

We wish to acknowledge and thank the Western Cape's PTC for their endorsement of the study. The authors would like to acknowledge the support of the MUE Committee, including J Louw, A Peters, P Steyn; and especially Zeenat Yusuf for her work with the pilot of the study and Renfred Joshua for designing the electronic data capture tool. We would also like to thank the pharmacists and doctors at healthcare facilities and the University of the Western Cape's School of Pharmacy students who assisted with the data collection.

## Author Contributions

**Conceptualization:** Renier Coetzee, Yasmina Johnson, Johan van Niekerk, Mosedi Namane.

**Data curation:** Renier Coetzee.

**Formal analysis:** Yasmina Johnson.

**Methodology:** Renier Coetzee, Yasmina Johnson, Mosedi Namane.

**Project administration:** Renier Coetzee.

**Resources:** Renier Coetzee.

**Supervision:** Renier Coetzee.

**Writing – original draft:** Renier Coetzee, Yasmina Johnson.

**Writing – review & editing:** Renier Coetzee, Johan van Niekerk, Mosedi Namane.

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
