## [Decision Letter · Decision Letter 0]

14 Jan 2020

PONE-D-19-33673

Amitriptyline prescribing in public sector healthcare facilities in the Western Cape, South Africa

PLOS ONE

Dear Prof Coetzee,

Thank you for submitting your manuscript to PLOS ONE. After careful consideration, we feel that it has merit but does not fully meet PLOS ONE’s publication criteria as it currently stands. Therefore, we invite you to submit a revised version of the manuscript that addresses the points raised during the review process.

We would appreciate receiving your revised manuscript by Feb 28 2020 11:59PM. To enhance the reproducibility of your results, we recommend that if applicable you deposit your laboratory protocols in protocols.io, where a protocol can be assigned its own identifier (DOI) such that it can be cited independently in the future. For instructions see: http://journals.plos.org/plosone/s/submission-guidelines#loc-laboratory-protocols

We look forward to receiving your revised manuscript.

Kind regards,

Vijayaprakash Suppiah, PhD

Academic Editor

PLOS ONE

Journal Requirements:

2. We noticed you have some minor occurrence(s) of overlapping text with the following previous publication(s), which needs to be addressed:

https://doi.org/10.1001/jama.286.22.2823

https://doi.org/10.1093/ajhp/56.5.433

http://www.healthconnect-intl.org/IRD_feb13.html

http://dx.doi.org/10.22270/jddt.v9i3.2649

In your revision ensure you cite all your sources (including your own works), and quote or rephrase any duplicated text outside the Methods section. Further consideration is dependent on these concerns being addressed.

3. In the ethics statement in the manuscript and in the online submission form, please provide additional information about the patient records/samples used in your retrospective study. Specifically, please ensure that you have discussed whether all data/samples were fully anonymized before you accessed them and/or whether the IRB or ethics committee waived the requirement for informed consent. If patients provided informed written consent to have data/samples from their medical records used in research, please include this information.

4. To comply with PLOS ONE submission guidelines, in your Methods section, please provide additional information regarding your statistical analyses. For more information on PLOS ONE's expectations for statistical reporting, please see https://journals.plos.org/plosone/s/submission-guidelines.#loc-statistical-reporting.

Reviewers' comments:

Reviewer's Responses to Questions

**Comments to the Author**

1. Is the manuscript technically sound, and do the data support the conclusions?

Reviewer #1: Yes

Reviewer #2: Yes

2. Has the statistical analysis been performed appropriately and rigorously? 

Reviewer #1: Yes

Reviewer #2: Yes

3. Have the authors made all data underlying the findings in their manuscript fully available?

Reviewer #1: Yes

Reviewer #2: No

4. Is the manuscript presented in an intelligible fashion and written in standard English?

Reviewer #1: No

Reviewer #2: Yes

5. Review Comments to the Author

Reviewer #1: I see that the data may not be free accessible, but I would presume it could be 'washed' of any identifying features?

As for the manuscript, the only comment I have is that it would be nice to see some of the statistics in a more compelling way. In particular, it is suggested that the big worry is the irrational prescribing for some subsets of the population. If the breakdown of the statistics was made for these populations within a table, some of the concerns/results would come through more clearly. For example, if Graph 2 was split for the elderly and for elderly males and females, that would really help to emphasise the point.

It would also be nice to get a bit more clarity on how the review of medical notes was done. Were there cases that were unclear? How were those addressed? Did each of the authors look through all of the notes to reach agreement, or was that chore split...

Reviewer #2: This is a useful review that uses standard methods . I would encourage publication after minor revisions as very few of these Medicine Use Evaluations are published in Low and Middle Income Countries. The work undertaken in this study could be replicated in many LMICs.

Specific Comments

Abstract The abstract includes the sentence "Amitriptyline is widely used in primary care in South Africa

and a cross-sectional study found that amitriptyline was prescribed potentially

inappropriately in 6.5% of patients." The reference to which this refers is I think reference 11. This was a study of "eldeley" patients. That should be made clear in the abstract.

Background Line 89. I suggest that the equivalent amount in USDollars be provided for the 3.5 million Rand to provide a context for non South African readers.

Methods I would like some more details of the "convenience sample". Clinical notes of outpatients in the public health sector of Western Cape could cover a broad range of facilities. Did the convenience sample of sixty four facilities come from all districts or was it limited to the Cape Town Metro region from which two of the four authors come. How many facilities with low amitryptiline facilities were excluded. What was the retrospective time period for which the records were reviewed?

Table 1 includes mention of Major Depressive Disorders. No abbreviation (MDD) for this term is provided. Later in the text on Lines 190 and 194 the abbreviation is used. I would suggest including the abbreviation the first time it is used.

Results Line 139 mentions that hospitals made up 20% of the sample. I would suggest repeating that only outpatient prescribing records were used. Hospital MUE's can be confused if inpatient and outpatient data are mixed.

Discussion I found the discussion very readable and interesting. One aspect that was not mentioned is how rare it is in LMICs for MUEs to be done for mental health medicines. Antibiotic use is frequently reviewed but mental health medicines are rarely studied. I was surprised at the relatively high rate of use according to guidelines. I was not surprised at the low dosage used for the treatment of depression. This has been reported elsewhere.

Conclusion. This is a useful study which has used robust methods that could be replicated in outer LMIC health systems. For that reason alone this paper should be published. The revisions I have suggested should be seen as clarifications.

6. PLOS authors have the option to publish the peer review history of their article (what does this mean?). If published, this will include your full peer review and any attached files.

Reviewer #1: Yes: Steven F Koch

Reviewer #2: No

---

## [Author Response · Author response to Decision Letter 0]

2 Mar 2020

Also see cover letter:

Comments addressed by authors:

This was noted and adjusted accordingly. The authors welcome further guidance from the lay-out editorial team.

2. We noticed you have some minor occurrence(s) of overlapping text with the following previous publication(s), which needs to be addressed:

https://doi.org/10.1001/jama.286.22.2823

https://doi.org/10.1093/ajhp/56.5.433

http://www.healthconnect-intl.org/IRD_feb13.html

http://dx.doi.org/10.22270/jddt.v9i3.2649

This was noted as an error and corrected. The article by Rajender et al was cited, while Zhan et al was found to reference the same authors as us, namely Wilcox et al (reference number 9). The last two links are websites that actually reference the World Health Organization documents listed in our reference list.

In your revision ensure you cite all your sources (including your own works), and quote or rephrase any duplicated text outside the Methods section. Further consideration is dependent on these concerns being addressed.

 3. In the ethics statement in the manuscript and in the online submission form, please provide additional information about the patient records/samples used in your retrospective study. Specifically, please ensure that you have discussed whether all data/samples were fully anonymized before you accessed them and/or whether the IRB or ethics committee waived the requirement for informed consent. If patients provided informed written consent to have data/samples from their medical records used in research, please include this information.

Additional information was provided under the sub-heading “Ethical Considerations”. Informed consent for patients were not required. Patient folders were reviewed retrospectively.

4. To comply with PLOS ONE submission guidelines, in your Methods section, please provide additional information regarding your statistical analyses. For more information on PLOS ONE's expectations for statistical reporting, please see https://journals.plos.org/plosone/s/submission-guidelines.#loc-statistical-reporting.

Noted. The necessary description was added in the method section.

This was noted. Data will be uploaded as supporting material. Statements updated as required.

5. Review Comments to the Author

Reviewer #1: I see that the data may not be free accessible, but I would presume it could be 'washed' of any identifying features?

Data will be made available as supporting information.

As for the manuscript, the only comment I have is that it would be nice to see some of the statistics in a more compelling way. In particular, it is suggested that the big worry is the irrational prescribing for some subsets of the population. If the breakdown of the statistics was made for these populations within a table, some of the concerns/results would come through more clearly. For example, if Graph 2 was split for the elderly and for elderly males and females, that would really help to emphasise the point.

For the purpose of the review we have complied with the request and provided a table (Table 3) containing the split. It should be noted that the numbers are then smaller and might not be as compelling. Our aim was not to answer our questions in this way, but rather to give a broad overview of use in a specific population. We welcome further comments and recommendations.

It would also be nice to get a bit more clarity on how the review of medical notes was done. Were there cases that were unclear? How were those addressed? Did each of the authors look through all of the notes to reach agreement, or was that chore split...

Addressed in the reviewed manuscript. Folders were reviewed independently and discrepancies were discussed with the main authors.

Reviewer #2: This is a useful review that uses standard methods . I would encourage publication after minor revisions as very few of these Medicine Use Evaluations are published in Low and Middle Income Countries. The work undertaken in this study could be replicated in many LMICs.

Specific Comments

Abstract The abstract includes the sentence "Amitriptyline is widely used in primary care in South Africa and a cross-sectional study found that amitriptyline was prescribed potentially

inappropriately in 6.5% of patients." The reference to which this refers is I think reference 11. This was a study of "eldeley" patients. That should be made clear in the abstract.

Background Line 89. 

Noted and corrected in the reviewed manuscript.

I suggest that the equivalent amount in USDollars be provided for the 3.5 million Rand to provide a context for non South African readers.

This was noted and corrected.

Methods I would like some more details of the "convenience sample". Clinical notes of outpatients in the public health sector of Western Cape could cover a broad range of facilities. Did the convenience sample of sixty four facilities come from all districts or was it limited to the Cape Town Metro region from which two of the four authors come. How many facilities with low amitryptiline facilities were excluded. What was the retrospective time period for which the records were reviewed?

We described the distribution of health facilities better and provided a breakdown of rural vs. urban facilities. Please see the amended text.

Table 1 includes mention of Major Depressive Disorders. No abbreviation (MDD) for this term is provided. Later in the text on Lines 190 and 194 the abbreviation is used. I would suggest including the abbreviation the first time it is used.

This was corrected at first mention. Major Depressive Disorders were written in full.

Results Line 139 mentions that hospitals made up 20% of the sample. I would suggest repeating that only outpatient prescribing records were used. Hospital MUE's can be confused if inpatient and outpatient data are mixed.

This was corrected to clearly describe outpatient use only. Inpatients were not part of the study.

Discussion I found the discussion very readable and interesting. One aspect that was not mentioned is how rare it is in LMICs for MUEs to be done for mental health medicines. Antibiotic use is frequently reviewed but mental health medicines are rarely studied. I was surprised at the relatively high rate of use according to guidelines. I was not surprised at the low dosage used for the treatment of depression. This has been reported elsewhere.

We note the above comment. Adding a comment on the rarity of utilization reviews done in LMIC were added in the background.

---

## [Decision Letter · Decision Letter 1]

30 Mar 2020

Amitriptyline prescribing in public sector healthcare facilities in the Western Cape, South Africa

PONE-D-19-33673R1

Dear Dr. Coetzee,

We are pleased to inform you that your manuscript has been judged scientifically suitable for publication and will be formally accepted for publication once it complies with all outstanding technical requirements.

With kind regards,

Vijayaprakash Suppiah, PhD

Academic Editor

PLOS ONE

Additional Editor Comments (optional):

Reviewers' comments:

Reviewer's Responses to Questions

**Comments to the Author**

1. If the authors have adequately addressed your comments raised in a previous round of review and you feel that this manuscript is now acceptable for publication, you may indicate that here to bypass the “Comments to the Author” section, enter your conflict of interest statement in the “Confidential to Editor” section, and submit your "Accept" recommendation.

Reviewer #1: All comments have been addressed

Reviewer #2: All comments have been addressed

2. Is the manuscript technically sound, and do the data support the conclusions?

Reviewer #1: Yes

Reviewer #2: Yes

3. Has the statistical analysis been performed appropriately and rigorously? 

Reviewer #1: Yes

Reviewer #2: Yes

4. Have the authors made all data underlying the findings in their manuscript fully available?

Reviewer #1: Yes

Reviewer #2: Yes

5. Is the manuscript presented in an intelligible fashion and written in standard English?

Reviewer #1: Yes

Reviewer #2: Yes

6. Review Comments to the Author

Reviewer #1: I only had rather minor comments the first time through, as did the other reviewer. In my view, the comments have been adequately addressed.

Reviewer #2: All reviewer comments have been addressed. I believe that this is a useful article to be published as there are very few MUE's of Mental health topics in LMICs.

7. PLOS authors have the option to publish the peer review history of their article (what does this mean?). If published, this will include your full peer review and any attached files.

Reviewer #1: Yes: Prof Steven F. Koch

Reviewer #2: Yes: Richard Laing

---

## [Editor Report · Acceptance letter]

6 Apr 2020

PONE-D-19-33673R1 

Amitriptyline prescribing in public sector healthcare facilities in the Western Cape, South Africa 

Dear Dr. Coetzee:

I am pleased to inform you that your manuscript has been deemed suitable for publication in PLOS ONE. Congratulations! Your manuscript is now with our production department. 

With kind regards,

on behalf of

Dr. Vijayaprakash Suppiah 

Academic Editor

PLOS ONE